# Protocol for the process evaluation of an intervention to improve antenatal smoking cessation support (MOHMQuit) in maternity services in New South Wales, Australia

Jo Longman,[1] Christine Paul [ORCID],[2] Aaron Cashmore,[3,4] Laura Twyman,[5] Larisa A J Barnes [ORCID],[1] Catherine Adams,[6] Billie Bonevski,[7] Andrew Milat,[4] Megan E Passey[8]

For numbered affiliations see end of article.

**Correspondence to**
Dr Larisa A J Barnes;
larisa.barnes@sydney.edu.au

## ABSTRACT

**Introduction** Smoking cessation in pregnancy remains a public health priority. Our team used the Behaviour Change Wheel to develop the Midwives and Obstetricians Helping Mothers to Quit smoking (MOHMQuit) intervention with health system, leader (including managers and educators) and clinician components. MOHMQuit addresses a critical evidence to practice gap in the provision of smoking cessation support in antenatal care. It involves nine maternity services in New South Wales in a cluster randomised stepped-wedge controlled trial of effectiveness. This paper describes the design and rationale for the process evaluation of MOHMQuit. The process evaluation aims to assess to what extent and how MOHMQuit is being implemented (acceptability; adoption/uptake; appropriateness; feasibility; fidelity; penetration and sustainability), and the context in which it is implemented, in order to support further refinement of MOHMQuit throughout the trial, and aid understanding and interpretation of the results of the trial.

**Methods and analysis** The process evaluation is an integral part of the stepped-wedge trial. Its design is underpinned by implementation science frameworks and adopts a mixed methods approach. Quantitative evidence from participating leaders and clinicians in our study will be used to produce individual and site-level descriptive statistics. Qualitative evidence of leaders' perceptions about the implementation will be collected using semistructured interviews and will be analysed descriptively within-site and thematically across the dataset. The process evaluation will also use publicly available data and observations from the research team implementing MOHMQuit, for example, training logs. These data will be synthesised to provide site-level as well as individual-level implementation outcomes.

**Ethics and dissemination** The study received ethical approval from the Population Health Services Research Ethics Committee for NSW, Australia (Reference 2021/ETH00887). Results will be communicated via the study's steering committee and will also be published in peer-reviewed journals and presented at conferences.

## STRENGTHS AND LIMITATIONS OF THIS STUDY

⇒ The process evaluation has been designed using implementation science frameworks.
⇒ The study uses multiple data sources. Qualitative and quantitative data will be collected independently from leaders and clinicians in each Midwives and Obstetricians Helping Mothers to Quit smoking (MOHMQuit) site as well as contextual and publicly available data, and observational data from the research team implementing MOHMQuit.
⇒ MOHMQuit is a complex intervention with many moving parts which interact with one another and the stakeholders involved. No process evaluation is able to collect data to understand all aspects of these interactions, particularly not in a 'real world' trial such as this one.

**Trial registration number** Australian New Zealand Trials Registry ACTRN12622000167763. https://www.australianclinicaltrials.gov.au/anzctr/trial/ACTRN12622000167763.

## INTRODUCTION

In 2020, 9.2% of mothers in Australia smoked tobacco at some point during their pregnancy.[1] Smoking in pregnancy is associated with a multitude of adverse outcomes for both mother and baby including preterm birth and low birthweight babies.[2–5] In Australia, smoking is the most common modifiable risk for adverse pregnancy and birth outcomes[6] and therefore supporting pregnant women to stop smoking remains a major public health concern and a priority for the New South Wales (NSW) Ministry of Health.[7–9] Clinical guidelines for NSW have existed for almost 20 years and recommend clinicians routinely provide evidence-based smoking cessation support (SCS) at all antenatal care visits for

| Antenatal Care clinicians | • Antenatal care midwives<br>• Aboriginal Health Workers (AHWs) - primary healthcare workers who ensure culturally safe maternity care in supporting Aboriginal and/or Torres Strait Islander women or women having an Aboriginal baby<br>• Obstetricians (staff specialists; Visiting Medical Officers with specialist obstetric training, Career Medical Officers) and obstetric registrars |
|---|---|
| Leaders | Maternity service leaders (those who support or supervise health professionals providing antenatal care), including:<br>• Clinical Midwifery Consultants<br>• Maternity Unit Managers<br>• Clinical Midwifery Educators<br>• Clinical Midwifery Specialists<br>• Antenatal clinic coordinators<br>• Obstetric leads |

**Figure 1** Description of key participant groups.

women who smoke or who have stopped smoking in this pregnancy.[10] Implementation of the guidelines shows room for improvement.[11–14] This fact, along with wider evidence that women want to stop smoking in pregnancy but some lack confidence to do so,[15] would value support from their clinicians[16] and a systematic review demonstrating that psychosocial interventions helps women to stop smoking,[17] led us to develop a theoretically underpinned intervention, Midwives and Obstetricians Helping Mothers to Quit smoking (MOHMQuit) to improve implementation of the NSW Guidelines.

### The MOHMQuit intervention

The MOHMQuit intervention has multiple components targeting different parts of a complex health system.[18] It is based on the '5As' of SCS: ask, advise, assess, assist and arrange follow-up, which has shown evidence of effectiveness for SCS.[19] MOHMQuit was developed using the Behaviour Change Wheel method.[20] It is an intervention built on local and international evidence identifying barriers and enablers for health systems, leaders and clinicians providing SCS.[21] It focuses on changing behaviours by targeting systems such as the electronic medical record system, leaders and clinicians (see figure 1 for further detail on what is meant by leaders and clinicians). For example, changing clinicians' behaviours so that they implement the guidelines by asking about smoking and discussing cessation at every antenatal visit, and assisting women by providing behavioural support such as discussing triggers for smoking, managing nicotine cravings and planning a quit attempt. The MOHMQuit trial is an implementation trial using a stepped-wedge design across five local health districts (LHDs) in NSW with diverse characteristics including organisational structure and staffing profiles.

The development of the MOHMQuit intervention and its support materials has been described in detail previously.[21 22] In brief, there are four main components (also referred to in the implementation science literature as 'implementation strategies'):

(1) Separate training events for maternity service leaders—half day, midwives and Aboriginal Health Workers—full day and obstetricians—2 hours. Midwifery educators also take part in the leaders' and midwives' training events as a 'train-the-trainer' model which

includes a comprehensive MOHMQuit training manual, and is central to the sustainability of the intervention.

(2) A number of MOHMQuit leadership processes and systems tools, for example, a report template for the electronic medical record system facilitating leaders' scrutiny of their services' SCS performance; a service audit tool for leaders.

(3) MOHMQuit written resources such as a booklet on 'Stopping smoking for you and your baby' for clinicians to use with women.

(4) A series of 11 short video clips for training and skills development to be used in a wide variety of settings, for example, at handover meetings.

Two months prior to the implementation starting in the first site, a day-long face-to-face gathering was held bringing together key decision-makers and clinicians from across the sites to ensure a shared awareness and understanding of MOHMQuit including its history and rationale, promote enthusiasm, motivation and engagement and establish shared understanding about roles and responsibilities.

At each site, 10 weeks prior to the intervention, the research team and the maternity service leaders will participate in a 'warm-up' meeting. While each site has a strong existing connection with MOHMQuit via the face-to-face day, and through the inclusion of partner investigators at each site, the warm-up meeting includes acknowledging and thanking those involved (which extend beyond the site partner investigators and include the antenatal clinic coordinator, the clinical midwifery educator and other leaders), generating enthusiasm, building momentum in the lead up to the implementation of MOHMQuit, and working through the logistics of implementation at each site. Two weeks prior to the intervention, a second meeting will be held which has a 'trouble-shooting' agenda and will also include detail of the research elements of MOHMQuit, for example, how and when outcome and process evaluation data from the site will be collected. Additional meetings are planned for 2 and 4 weeks post intervention, to maintain momentum and explore any unresolved issues in the ongoing implementation of MOHMQuit. A MOHMQuit Community of Practice will be established which each site can join following implementation. The Community of Practice will offer a regular forum for sharing and supporting other clinicians and leaders in continuing to implement MOHMQuit and is one of several sustainability features of MOHMQuit. Finally, three and a half months after implementation, each site will receive feedback from brief interviews with women about the SCS they received during their antenatal care. They will continue to receive these reports quarterly until the end of the trial.

MOHMQuit is currently being trialled in a multisite cluster randomised stepped-wedge effectiveness trial in nine sites in publicly funded maternity services in NSW, Australia.[23] Implementation is planned to take place over a 13-month timeframe. Unlike many earlier interventions aimed at improving SCS,[24] MOHMQuit is built

on implementation science frameworks and is specific to the public maternity service setting. The trial will assess the intervention outcomes. The primary intervention outcome is smoking cessation, and secondary intervention outcomes include changes to clinicians' knowledge, skills, confidence and behaviour in providing SCS and test the 'mechanisms of action'[25–27] by which each of the components affect intervention outcomes and moderators of their impact in this framework-driven approach.[23] Cost-effectiveness will be assessed in an economic evaluation.[28] The trial will also assess key implementation outcomes (assessing how MOHMQuit was implemented) primarily based on Proctor *et al*'s implementation science framework[29] in a detailed process evaluation. The process evaluation will complement the assessment of the MOHMQuit intervention outcomes. Conducting process evaluation alongside effectiveness trials in this way is recommended.[30 31]

### Aims of the MOHMQuit process evaluation

Process evaluations explore how an intervention is implemented. They assess three aspects: (1) how and to what extent the intervention was implemented; (2) the 'mechanisms of impact' that is, how the intervention components and participants' interactions with these components effected changes in behaviour; and (3) the context in which the intervention was implemented.[32] We anticipate that the process evaluation will contribute formatively by providing feedback that may further refine the intervention. This is particularly useful in a stepped-wedge trial design where each site joins the trial sequentially, and acceptable as long as the changes made to components retain the integrity of the function they were meant to perform in the original intervention design.[33 34] The summative use of process evaluation is in providing insight into the mechanisms through which the intervention outcomes (the primary intervention outcome being pregnant women stopping smoking) were achieved or not, and therefore it will contribute to understanding and interpreting the results of the effectiveness trial.[35] Without this insight effective, and ineffective, aspects of the intervention may not be understood and this has implications for the scale-up of an intervention such as MOHMQuit. In this way, the process evaluation will maximise the knowledge gained throughout the trial and describe the most effective delivery processes for the MOHMQuit intervention. The aim of this protocol paper is to describe the process evaluation planned as an integral part of the MOHMQuit trial.

### METHODS AND ANALYSIS
### Overall design and objectives of the process evaluation

The design for the process evaluation began with the implementation outcomes defined by Enola Proctor and team in order to facilitate an understanding of the various dimensions of the implementation: acceptability; adoption/uptake; appropriateness; feasibility; fidelity;

penetration and sustainability (and sustainment).[29] Implementation outcomes are "…the effects of deliberate and purposive actions to implement new…practices".[29] The Proctor implementation outcomes generally map on to other well-used frameworks such as the Reach, Efficacy, Adoption, Implementation, Maintenance (RE-AIM) framework[31] but 'Reach' from the RE-AIM framework was specifically added into the design as 'Reach' captures the number of clinicians and leaders invited to and taking part in the trial. Two other frameworks informed the implementation outcomes of interest: Sekhon[36] for acceptability, and Rogers[37] for sustainability, appropriateness and feasibility; and Moore[35] and Fernandez'[38] work guided exploration of mechanisms of impact and how context affected implementation. The context in which the intervention was implemented will also be assessed. Context is variously defined[39] but here contextual features are conceived of broadly as those influencing the delivery of the intervention and include the engagement of leaders and the organisational setting and culture of the service in which the intervention is implemented.[40]

Important features of the process of implementing MOHMQuit were discussed and agreed with a process evaluation working group of the project's steering committee (a key governance committee of the project and constituted of research academics, policy-makers, managers and leaders[21]). Subsequently, instruments were developed which encompassed both individual and service-level data collection. Decisions were made regarding the specific foci of the process evaluation, acknowledging that "Process evaluations cannot expect to provide answers to all of the uncertainties of a complex intervention. It is generally better to answer the most important questions well than to try to answer too many questions and do so unsatisfactorily".[35]

With that in mind, a focus on fidelity; adoption/uptake; penetration; reach, sustainability and context was agreed. In part, these foci were based on learning from the feasibility and acceptability trial of MOHMQuit.[21] In addition, the short duration of the trial (the time from implementation at the first site to the end of data collection, excluding the wash-out period, is 24 months and from the final site, only 8 months) would make sustainment challenging to measure. Sustainability is, however, included in the evaluation. Sustainment is "…the continued use of a practice that is the target of the implementation, whereas sustainability addresses whether the factors are in place to promote the ongoing use".[41]

The process evaluation has three interrelated objectives; to, at both the individual and site levels, assess

1. To what extent MOHMQuit was implemented—measured quantitatively focusing on the implementation outcomes of adoption, fidelity, penetration, reach and sustainability, and will also involve qualitative measures (interviews with leaders).
2. How changes in behaviour were effected (the mechanisms of impact)—measured quantitatively focusing on the implementation outcomes of acceptability, ap-

propriateness and feasibility, and a more nuanced understanding of this from leaders' perspectives in qualitative interviews.

3. The impact of context (moderators) on the implementation of MOHMQuit. A moderator is a factor that will strengthen or lessen the influence of a strategy to implement MOHMQuit.[26] We anticipate a number of moderators will be an important part of the context for MOHMQuit implementation, as well as intervention outcomes, affecting the relationship between the implementation outcomes, for example, reach, and the implementation of MOHMQuit. The moderators measured include

   a. Leadership
      i. Leaders self-assessment of their leadership for implementation at 3 months using the Implementation Leadership Scale.[42]
      ii. Clinicians questionnaires at 6 months which include the Leadership Engagement Scale.[38]
   b. Implementation climate
      i. Clinician questionnaires at 6 months which include the Implementation Climate Scale.[38]
   c. Service size
   d. Smoking prevalence among pregnant women birthing at that site.
   e. Other demands on leaders/service, for example, new SCS policies and training or accreditation.

In summary, we speculate that the impact of the context on the implementation outcomes could be as follows:

Leadership and implementation climate—impacting on all outcomes.

Service size, smoking prevalence and models of care—impacting on adoption, appropriateness, feasibility, penetration and sustainability.

Other demands on leaders—impacting on implementation in terms of adoption, fidelity, penetration and sustainability.

Figure 2 summarises our speculation about which of each of the context elements might impact on each of the implementation outcomes; for example, we anticipate leadership will impact on all of the implementation outcomes.

### Recruitment and consent

The LHDs (which manage public hospitals and provide healthcare services in a defined geographical area) in NSW with relatively high rates of smoking in pregnancy were approached to participate in the MOHMQuit trial.

| Context measures | Implementation outcomes | | | | | | | |
|---|---|---|---|---|---|---|---|---|
| | Acceptability | Adoption | Appropriateness | Feasibility | Fidelity | Penetration | Sustainability | Reach |
| Leadership | ■ | ■ | ■ | ■ | ■ | ■ | ■ | ■ |
| Implementation climate | ■ | ■ | ■ | ■ | ■ | ■ | ■ | ■ |
| Service size | | ■ | ■ | ■ | | ■ | ■ | |
| Smoking prevalence | | ■ | ■ | ■ | | ■ | ■ | |
| Models of care | | ■ | ■ | ■ | | ■ | ■ | |
| Other demands on leaders | | ■ | | ■ | ■ | ■ | ■ | |

**Figure 2** Speculating which context elements may impact on each of the implementation outcomes.

There are 15 LHDs in total, 7 with high smoking rates in pregnancy were invited and 5 agreed to participate in the trial. Between them, they selected nine maternity services (sites) to take part. The senior midwives and lead obstetricians from these five LHDs were partner investigators in a partnership grant application subsequently awarded by Australia's National Health and Medical Research Council and so their involvement with the project substantially precedes the implementation trial of MOHMQuit.

Individual service leaders and clinicians in each of the nine sites will be provided with a participant information sheet and those who agree to participate in the research will be asked to sign a written consent form indicating their consent to take part in data collection. This consent applies to data collection to measure the implementation outcomes and context as well as the intervention outcomes.

### Process evaluation data collection

The process evaluation will adopt a mixed methods approach, collecting quantitative evidence from questionnaires and qualitative evidence of leaders' perceptions of how MOHMQuit may have changed behaviour (where it was perceived to have done so) from semistructured interviews. Data will be collected by the research team independently from each of the nine MOHMQuit sites. Study-specific questionnaires will be used to collect implementation outcome data from leaders and clinicians at each site at various time points: immediately following training, 3 months after the training and 6 months after the training. To minimise participant burden, the questionnaires will also collect the data required to measure the intervention outcomes.

Qualitative data will be collected using semistructured interviews with leaders 6 months after the training at each site. The interviews have three key purposes. First, interviews will collect data on the components of MOHMQuit which have been implemented in the 6 months following the MOHMQuit training (uptake), for example, use of the report template for the electronic medical record system facilitating leaders' scrutiny of their services' SCS performance for feedback and continuous improvement, or MOHMQuit training delivered by the service themselves using the train-the-trainer manual. Second, they will collect data to support the calculation of an implementation cost as part of the detailed economic evaluation of MOHMQuit, the subject of a separate paper,[28] by recording how much time leaders' assess they spend implementing those components of MOHMQuit. Finally, they will collect data which will enhance the contextual information collected by the research team by eliciting leaders' perspectives of the enablers and barriers of the implementation of MOHMQuit and what might be improved with regard to it. Interviews will be conducted using the Teams platform, recorded and transcribed. They will be guided by an interview schedule driven by the implementation outcomes and the contextual factors that

supported or hindered implementation and any adaptations made to the intervention. The semistructured nature of the interviews will allow for flexibility in questioning and expansion on responses.

Data collection from leaders and antenatal care clinicians will be as follows:

### Leaders
► An online questionnaire to all leaders 3 months after the training at each site regardless of whether they attended MOHMQuit training (anticipated numbers of leaders who will be invited approximately 55).
► A semistructured one-to-one telephone interview 6 months after the training with the midwifery partner investigator and one to two other leaders at each site.

### Antenatal care clinicians
► A paper questionnaire immediately following the training at each site to participants who attended training (anticipated numbers of participants who will be invited approximately 250).
► An online questionnaire to all antenatal care clinicians and AHWs 6 months after the training at each site regardless of whether they attended MOHMQuit training (anticipated numbers of participants who will be invited approximately 300).

In addition, attendance and fidelity information (which aspects of the training were delivered) will be kept by the research team during each training event and the attendance and engagement at various meetings that are components of MOHMQuit. The additional data collection includes
► Training logs—to calculate proportion attended at each training event (attendance/invited).
► A 'fidelity checklist' of which elements of the training were covered during each training event.
► Attendance and notes from 10-week warm-up meetings.
► Attendance and notes from 2-week warm-up meetings.
► Attendance and notes from 2-week follow-up meetings.
► Attendance and notes from 4-week follow-up meetings.
► Attendance and notes from monthly community of practice meetings.

For each site, a 'context table' will be completed by the research team using publicly available sources and with input from partner investigators at each site (box 1).

We anticipate that the data collection itself may have the beneficial sustainability effect of reminding leaders and clinicians about MOHMQuit and possibly prompting renewed attention and/or commitment to it.

Table 1 provides detail of working definitions and how each of the implementation outcomes and contextual features will be measured at which timepoints, using which instruments with whom, and which strategies (components of the MOHMQuit intervention) are aimed to maximise the implementation outcomes. Further detail is available in online supplemental table 1.

---

| **Box 1  Key contextual information collected for each site** |
| :--- |
| ⇒ Number of births at site 2020 |
| ⇒ Smoking prevalence 2020 |
| ⇒ Performance against the NSW Ministry of Health's performance indicator of antenatal smoking |
| ⇒ Safer Baby Bundle at site?* (Yes/No) |
| ⇒ Preparation and training for new NSW Maternity Care Policy (RSVP - Reducing the effects of Smoking and Vaping on Pregnancy and newborn outcomes)† overlaps with MOHMQuit timing? (Yes/No) |
| ⇒ Other smoking cessation support initiatives running at the site? (Yes/No) |
| ⇒ Accreditation for Quality Improvement going on concurrent with MOHMQuit? (Yes/No) |
| ⇒ Leadership structure at the site |
| ⇒ Models of care offered and proportion of women at booking and at birth for each model |
| ⇒ Other, for example, external events like disasters, vacant posts |

*Safer Baby Bundle is a multicomponent intervention in maternity service which aims to reduce the number of preventable stillbirths.
†The RSVP policy is a policy directive establishing minimum requirements for health services to provide evidence-based smoking cessation support to women before during and after pregnancy. The RSVP Policy was released 14 October 2022.

### Patient and public involvement
As this is an implementation science trial, our partners in identifying the need for the study and in its design and implementation were health service clinicians, leaders and policy-makers.[21] Patients were not involved in designing or implementing the research, but are participants in the trial[23] but not in the process evaluation.

### Data analysis
We will assess each of the implementation outcomes (table 1) for each site, including assessing variation across the nine sites. At this stage, it is not possible to definitively describe which of the implementation outcomes our analyses will be focused on as that will depend on the variation in implementation outcomes across sites. For example, if there is little variation in fidelity, it will not help explain the MOHMQuit (intervention) outcomes. However, where appropriate, descriptive statistics (measures of central tendency, SD and proportions) will be produced using data from questionnaire responses from clinicians and leaders to summarise quantitative results by participant and by site.

Analyses for the moderators will include calculation of a measure of central tendency, for the leadership[42] subscales for each participant. There are four subscales: the proactive subscale, the knowledgeable subscale, the supportive subscale and the perseverant subscale. A measure of central tendency for each set of items that load onto the relevant subscale will be calculated for each subscale. A measure of central tendency of the scale scores will be calculated which will provide a total score for the Implementation

**Table 1** Implementation outcomes, definitions, strategies for maximising implementation outcomes, frameworks used and measurement items

| Implementation outcome | Strategies used to maximise implementation outcomes | Frameworks used | Data collection instruments, timing, participants |
|---|---|---|---|
| Adoption/uptake (intention or action to try to employ MOHMQuit) | ▶ Warm-up and follow-up meetings<br>▶ Community of practice | Proctor[29]<br>RE-AIM[31]<br>(adoption) | ▶ Attendance at warm-up and follow-up meetings<br>▶ 3 months post training—questionnaire with leaders<br>▶ 6 months post training—questionnaire with clinicians<br>▶ 6 months post training—interview with leaders<br>▶ Community of practice peer support meetings attendance |
| Fidelity (delivered as intended in the Protocol,[23] adherence) | ▶ Warm-up and follow-up meetings<br>▶ Consistency in the team delivering MOHMQuit training at each site in the first instance<br>▶ Clear plans and materials for content of training | Proctor[29]<br>RE-AIM[31]<br>(implementation) | ▶ Attendance at warm-up and follow-up meetings<br>▶ Training logs of expected and actual attendance at training of leaders and clinicians<br>▶ Fidelity record (which aspects of the planned training were actually delivered)<br>▶ 6 months post training—interview with leaders |
| Penetration (degree of integration of MOHMQuit practices within the service) | ▶ Involving leaders in the training for clinicians for a whole-of-service approach<br>▶ MOHMQuit leadership components include repeated audit and feedback plus action planning; developing and implementing a clinical pathway for SCS; and the development and maintenance of SCS 'champions' within each service<br>▶ Train-the-trainer model an integral part of the intervention | Proctor[29]<br>RE-AIM[31]<br>(adoption) | ▶ 3 months post training—questionnaire with leaders<br>▶ 6 months post training—interview with leaders |
| Reach (did MOHMQuit include everyone that it aimed to?) | ▶ 10-week warm-up meetings to allow time for planning and rostering<br>▶ The train-the-trainer model as an integral part of the intervention to support participation of all relevant existing and new staff | RE-AIM[35] | ▶ Training logs of expected and actual attendance at training of leaders and clinicians recorded at the time of training<br>▶ 3 months post training—questionnaire with leaders<br>▶ 6 months post training—interview with leaders |
| Sustainability (factors promoting ongoing use of MOHMQuit) | ▶ MOHMQuit leadership components include repeated audit and feedback plus action planning; developing and implementing a clinical pathway for SCS; and development and maintenance of SCS 'champions' within each service<br>▶ Train-the-trainer model an integral part of the intervention<br>▶ The Community of practice | Proctor[29]<br>RE-AIM[31]<br>(maintenance)<br>Rogers[37] | ▶ 6 months post training—questionnaire with clinicians<br>▶ 6 months post training—interview with leaders<br>▶ Community of practice peer support attendance data |

Continued

**Table 1** Continued

| Implementation outcome | Strategies used to maximise implementation outcomes | Frameworks used | Data collection instruments, timing, participants |
|---|---|---|---|
| Acceptability (how palatable is MOHMQuit to clinicians and leaders?) | ▶ Comprehensive systematic design of MOHMQuit using the Behaviour Change Wheel with input from clinicians and leaders[21 22]<br>▶ Feasibility and acceptability trial with subsequent minor amendments to the intervention[21]<br>▶ 10-week warm-up includes the history of MOHMQuit so leaders are reassured about its quality, relevance and acceptability | Proctor[29] Sekhon[36] | ▶ Immediately post training—questionnaire with clinicians<br>▶ 3 months post training—questionnaire with leaders<br>▶ 6 months post training—questionnaire with clinicians<br>▶ 6 months post training—interview with leaders |
| Appropriateness (perceived fit or relevance of MOHMQuit with the service) | ▶ Comprehensive and systematic design of MOHMQuit using the Behaviour Change Wheel integrating input from clinicians and leaders[21 22]<br>▶ Feasibility and acceptability trial with subsequent minor amendments to the intervention[21]<br>▶ 10-week warm-up includes the history of MOHMQuit so leaders are reassured about its quality, relevance and acceptability | Proctor[29] Rogers[37] | ▶ 6 months post training—interview with leaders |
| Feasibility (actual fit—the extent to which MOHMQuit can be integrated into usual care in a service) | ▶ Comprehensive and systematic design of MOHMQuit using the Behaviour Change Wheel integrating input from clinicians and leaders[21 22]<br>▶ Feasibility and acceptability trial with subsequent minor amendments to the intervention[21]<br>▶ 10-week warm-up includes the history of MOHMQuit so leaders are reassured about its quality, relevance and acceptability | Proctor[29] Rogers[37] | ▶ 3 months post training—questionnaire with leaders<br>▶ 6 months post training—interview with leaders |
| HOW behaviour was changed | | Moore[35] | ▶ 6 months post training—interview with leaders |
| **HOW context affected implementation** | **▶ Commitment of maternity service leaders in the research as Partner Investigators and members of MOHMQuit Steering Committee and various working groups**<br>**▶ Warm-up meetings and follow-up meetings**<br>**▶ Community of practice** | **Fernandez[38]** | **▶ Key contextual information (box 1) completed by research team during the implementation**<br>**▶ 3 months post training—questionnaire with leaders**<br>**▶ 6 months post training—questionnaire with clinicians**<br>**▶ 6 months post training—interview with leaders** |

Bold typeface indicates outcomes that will be the focus of the process evaluation.
Implementation cost is not included in table 1 as a detailed economic evaluation of MOHMQuit is taking place and is the subject of a separate paper.[28] Data to contribute to the economic evaluation will be collected as part of the semistructured interview with leaders.
MOHMQuit, Midwives and Obstetricians Helping Mothers to Quit smoking; RE-AIM, Reach, Efficacy, Adoption, Implementation, Maintenance; SCS, smoking cessation support.

Leadership Scale.[42] In addition, scores will be aggregated to provide a site-level score. We do not anticipate adding these results, or any of the data from box 1 to any model but they will help constitute a broader assessment of the context for implementation to contribute to understanding of in which sites, and how, MOHMQuit was effective.

Qualitative data from semistructured interviews with leaders will be analysed descriptively to explore perspectives of uptake by site and thematically across

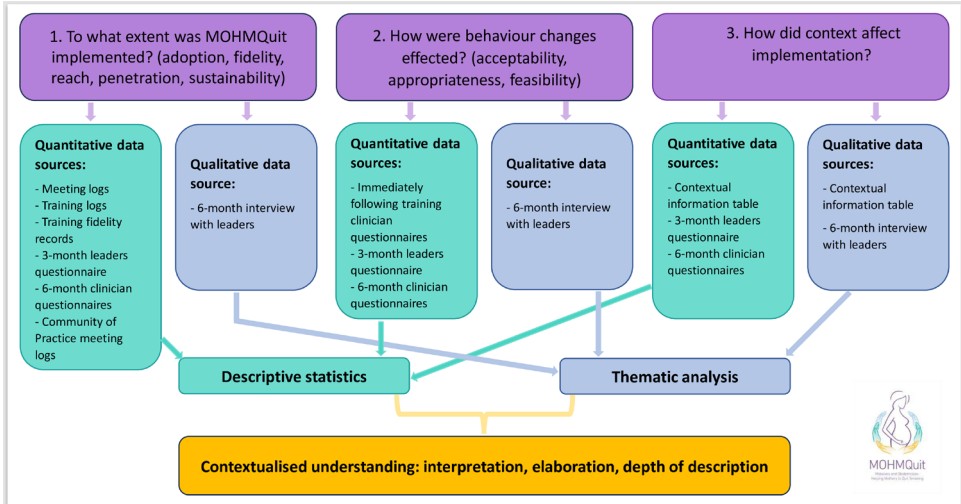

**Figure 3** Mixed methods approach to data collection and analysis.

all sites regarding the enablers, barriers and how implementation of MOHMQuit might be improved. Thematic analysis will follow the steps outlined by Braun and Clarke: data familiarisation; initial generation of codes; development of themes by collating codes and then reviewing the raw data again to check the material sits together coherently as a theme; defining each theme and how themes work together to tell the overall story of the data.[43]

Data from multiple sources will facilitate triangulation, for example, collecting data about acceptability from quantitative data (post-training questionnaires from clinicians and questionnaires at 6 months from all clinicians) along with qualitative interviews with leaders from each site. This mixed methods approach will broaden and deepen understanding of the results of the trial. The key findings will be presented in an integrated way using a side-by-side joint display table,[44] each source being given equal weight. Figure 3 describes this visually.

### Ethical considerations and dissemination

The process evaluation received ethical approval from the NSW Population Health Services Research Ethics Committee (Reference Number 2021/ETH00887) on 23 July 2021. Results of the process evaluation will be written up for publication in peer-reviewed journals and presented at conferences. The process evaluation will perform a formative function facilitated by the stepped-wedge design with sites receiving the intervention in a staggered implementation, allowing for further polishing of the intervention as the trial proceeds. The process evaluation will also provide contextual information to elucidate the findings of the trial in terms of how MOHMQuit may have been effective in some sites but not in others. This understanding is critical in relation to rolling out MOHM-Quit across NSW should the intervention prove to be effective.

### Trial registration number

The MOHMQuit trial is registered with ANZCTR (www.anzctr.org.au): ACTRN12622000167763.

### DISCUSSION

Implementation science is the study of approaches that support the systematic uptake of research findings into 'usual care'.[45] In cases where there is an urgent need for behaviour change and clear evidence to practice gap, such as with SCS in antenatal care, implementation science provides a framework for examining an intervention such as MOHMQuit. This paper describes the mixed-method design and underpinning frameworks for the process evaluation of MOHMQuit as part of an implementation science study. MOHMQuit is a complex multi-component intervention designed using the Behaviour Change Wheel.[20] It aims to change the behaviour of antenatal care providers to improve the support provided to women to stop smoking in pregnancy. MOHMQuit is being implemented in a stepped-wedge effectiveness trial across nine publicly funded maternity services in NSW.[23]

The process evaluation will facilitate the ongoing refinement of MOHMQuit and will provide an assessment of the extent to which MOHMQuit was implemented, what the mechanisms of impact were and what the context of implementation was, and how it affected the implementation of MOHMQuit. It will also inform other components of the study, for example, contributing data to support costing of MOHMQuit for the economic evaluation. We anticipate that the findings from the process evaluation will contextualise and aid understanding of our trial results, and may support the further implementation of MOHMQuit in NSW. For example, if it transpires that implementation leadership is more evident in those sites where MOHMQuit was shown to be particularly effective, the scale-up would need to include a focus on *implementation leadership* and on implementing the leadership components of the intervention. Our process evaluation

will also contribute knowledge about the implementation of stepped-wedge trials which may be useful to others in the future. While we have described our intended approach to evaluating the implementation of MOHM-Quit, we have also included flexibility of approach in recognition of unanticipated implementation factors that may surface.[40]

Smoking in pregnancy is an ongoing public health challenge and represents a considerable gap between the evidence for SCS and practice. Providing a broader understanding of how MOHMQuit was or was not effective will be key to its potential future roll-out/scale-up.

## Empirical testing of the theory

Implementation science is a relatively new academic endeavour and this process evaluation has the potential to contribute to a growing body of evidence of approaches to implementing comprehensive stepped-wedge trial designs that are inclusive of process evaluation.

## Strengths and limitations

The process evaluation has been designed using implementation science frameworks and explores the implementation of MOHMQuit, a thorough and theoretically underpinned intervention and trial design.[21] The results of the trial will provide further evidence for the effectiveness, or otherwise, of this theoretically driven approach. The mixed methods approach in the process evaluation includes qualitative and quantitative data collection from a wide range of leaders and clinicians in each MOHMQuit site, some of whom will not have directly participated in the MOHMQuit training, as well as publicly available data and observational data from the research team implementing MOHMQuit. This approach has the potential to produce findings that have depth and nuance and will aid understanding of the trial findings. However, MOHM-Quit is a complex intervention with many moving parts which interact with one another, and the stakeholders involved. No process evaluation is able to collect data to understand all aspects of these interactions. In addition, the MOHMQuit trial is a 'real world' trial. This has strengths in producing findings that can be confidently understood as realistic; however, it also produces many challenges including the potential impact of new policies and procedures, staffing issues, and so on, many of which we have aimed to record as part of the process evaluation but some of which we are likely to have missed. This may compromise our capacity to fully understand and accurately interpret the intervention outcomes.

## Trial status

Recruitment for the trial is underway. Process evaluation and data collection commenced in March 2023 and will conclude in May 2024.

## Author affiliations
[1] University Centre for Rural Health, The University of Sydney, Sydney, New South Wales, Australia

[2] School of Medicine and Public Health, University of Newcastle, Callaghan, New South Wales, Australia
[3] Centre for Epidemiology and Evidence, NSW Health, St Leonards, New South Wales, Australia
[4] School of Public Health, Faculty of Medicine and Health, The University of Sydney, Sydney, New South Wales, Australia
[5] Tobacco Control Unit, Cancer Prevention Division, Cancer Council NSW, Woolloomooloo, New South Wales, Australia
[6] Clinical Excellence Commission, NSW Health, St Leonards, New South Wales, Australia
[7] Flinders Health and Medical Research Institute, College of Medicine and Public Health, Flinders University, Bedford Park, Adelaide, South Australia, Australia
[8] Daffodil Centre and the University Centre for Rural Health, Faculty of Medicine and Health, The University of Sydney, Sydney, New South Wales, Australia

**Acknowledgements** This paper is submitted on behalf of the MOHMQuit Trial team, including all chief investigators, partner investigators and associate investigators, and co-researchers and site leads at each of the MOHMQuit sites. In addition to the named authors, the team includes Dheya Al Mashat (NSW Health), Dianne Avery (NSW Health), Elizabeth Best (NSW Ministry of Health), Alecia Brooks (Cancer Council NSW), Rashna Chinoy (NSW Health), Justine Elliot (NSW Health), Jacinta Felsch (NSW Health), Mohamed Foda (NSW Health), Sandra Forde (NSW Health), Tara Farrugia (NSW Health), Tracey Greenberg (Alcohol and Drug Service, St Vincent's Hospital Sydney), Jane Griffith (NSW Health), Madeline Hubbard (NSW Health), Damien McCaul (NSW Ministry of Health), James McLennan (Alcohol and Drug Service, St Vincent's Hospital Sydney), Kate Reakes (Cancer Institute NSW), Virginia Stulz (NSW Health and Western Sydney University), Tracey Zakazakaarcher (NSW Health), and Lou Atkins (University College London) who provided excellent early guidance on the process evaluation design.

**Contributors** The process evaluation was conceived and designed by all authors. The first draft of the paper was written by JL with input from MP and CP before receiving input from all other authors. All authors have read and approved the final manuscript.

**Funding** This work is currently supported by a partnership grant from the NHMRC (GNT1185261). Previously, NHMRC funding (GNT1072213) and the Cancer Institute NSW funding (13/ECF/1-11) supported the developmental work of MOHMQuit. MP was also supported during this time by a fellowship grant from the NHMRC (GNT1159601).

**Competing interests** None declared.

**Patient and public involvement** Patients and/or the public were not involved in the design, or conduct, or reporting, or dissemination plans of this research.

**Patient consent for publication** Not applicable.

**Provenance and peer review** Not commissioned; externally peer reviewed.

**ORCID iDs**
Christine Paul http://orcid.org/0000-0002-0504-5246
Larisa A J Barnes http://orcid.org/0000-0002-9847-775X

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
