## [Reviewer comments · BMJ Open]

ARTICLE DETAILS

TITLE (PROVISIONAL)	Protocol for the process evaluation of an intervention to improve antenatal smoking cessation support (MOHMQuit) in maternity services in New South Wales, Australia
AUTHORS	Longman, Jo; Paul, Christine; Cashmore, Aaron; Twyman, Laura; Barnes, Larisa; Adams, Catherine; Bonevski, Billie; Milat, Andrew; Passey, Megan

VERSION 1 – REVIEW

REVIEWER	Felix Naughton University of East Anglia Faculty of Medicine and Health Sciences, School of Health Sciences
REVIEW RETURNED	12-Dec-2023

GENERAL COMMENTS	This is a thoughtful and thorough process evaluation protocol. I am not an implementation scientist but as far as I can tell, the authors use key frameworks and tools from implementation science appropriately. A few minor observations below: Spell out acronym of NSW in abstract. I wonder if the authors are using the term 'implementation' somewhat differently to some other people might? In the introduction it says "Process evaluations explore how an intervention is implemented" which includes "the 'mechanisms of impact' i.e. how the intervention components and participants' interactions with these components effected changes in behaviour." Some, like me, would use the term 'implementation' to relate to the real-world delivery of an intervention, usually outside of an efficacy/effectiveness evaluation. Many process evaluations of trials do not investigate implementation in this meaning of the term. Given how differently this term can be used, I would recommend the authors define it early on in the protocol. It is broken down in the beginning of the 'Overall design...' section, though I feel it would be helpful if defined earlier. Could the authors clarify what behaviour is being referred to for "2. How changes in behaviour were effected...". Presumably this is focusing on the health organisations/leaders/clinicians behaviour, but which behaviours? The list of outcomes focused on ("acceptability, appropriateness and feasibility") seems to leave quite a gap for other potential mechanisms of action (adherence, uptake/reach, and individual/psychological factors such as motivation, attitude/cognitive change, self-efficacy etc.)? I appreciate the authors highlight how not all process evaluation questions can be answered but some of these seem pretty critical. Data analysis – I suggest the authors remain open to other
---

	measures of central tendency other than the mean in case distributions of scores are not normal – the median is usually preferable (if data is normally distributed it will align with the mean, if not, it will be a more valid indicator of central tendency). The qualitative analysis section was brief and did not really describe how themes would be generated. Figure 3 needs some re-formatting (probably the process of converting to pdf), though is a helpful overview. The references need some re-formatting too as some of the organisation names/authors haven't come across properly (I have the same issues with my reference management software and have to hand correct them!). After the reference section some of the tables are repeated but without titles.
--	--

REVIEWER	Annariina Koivu Tampere Universities
REVIEW RETURNED	19-Dec-2023

GENERAL COMMENTS	Dear Authors, Thank you for this interesting protocol article on a process evaluation of a psychosocial intervention to improve antenatal smoking cessation support. I think there is room for theory-driven, pragmatic evaluation designs, that can improve transferability, adaptation and scale-up of an intervention. The protocol for the most part is well-written and sufficiently detailed. However, I have several minor comments, that I have made directly to the manuscript using comments function. I have attached the commented version of the manuscript below. I recommend that article will be accepted with minor revisions. The reviewer provided a marked copy with additional comments. Please contact the publisher for full details.
--

VERSION 1 – AUTHOR RESPONSE

Reviewer: 1

Dr. Felix Naughton, University of East Anglia Faculty of Medicine and Health Sciences

Reviewers' comment	Response (page numbers from the revised word document)
Spell out acronym of NSW in abstract.	Thank you – amended.

I wonder if the authors are using the term ‘implementation’ somewhat differently to some other people might? In the introduction it says “Process evaluations explore how an intervention is implemented” which includes “the ‘mechanisms of impact’ i.e. how the intervention components and participants’ interactions with these components effected changes in behaviour.” Some, like me, would use the term ‘implementation’ to relate to the real-world delivery of an intervention, usually outside of an efficacy/effectiveness evaluation. Many process evaluations of trials do not investigate implementation in this meaning of the term. Given how differently this term can be used, I would recommend the authors define it early on in the protocol. It is broken down in the beginning of the ‘Overall design...’ section, though I feel it would be helpful if defined earlier.	We have amended the text in the Introduction (last paragraph on p5) to highlight that we are assessing implementation outcomes from Implementation Science frameworks (as defined by Proctor et al, and others) in the process evaluation, and that these are distinct from the intervention outcomes: The trial will also assess key implementation outcomes (assessing how MOHMQuit was implemented) primarily based on Proctor et al’s implementation science framework¹ in a detailed process evaluation. The process evaluation will complement the assessment of the MOHMQuit intervention outcomes.
Could the authors clarify what behaviour is being referred to for “2. How changes in behaviour were effected...”. Presumably this is focusing on the health organisations/leaders/clinicians behaviour, but which behaviours? The list of outcomes focused on (“acceptability, appropriateness and feasibility”) seems to leave quite a gap for other potential mechanisms of action (adherence, uptake/reach, and individual/psychological factors such as motivation, attitude/cognitive change, self-efficacy etc.)? I appreciate the authors highlight how not all process evaluation questions can be answered but some of these seem pretty critical.	We have added some examples of what behaviour we mean early in the Introduction – essentially this is clinicians actually enacting the Guidelines: For example, changing clinicians’ behaviours so that they implement the Guidelines by asking about smoking and discussing cessation at every antenatal visit, and assisting women by providing behavioural support such as discussing triggers for smoking, managing nicotine cravings, and planning a quit attempt. (p4 second paragraph) Table 2 details aspects of feasibility and acceptability that will pick up on some of the factors this reviewer lists such as motivation, attitude, self-efficacy. For example in the 6-month questionnaire for clinicians there are questions about how confident clinicians perceive themselves to be in providing SCS 6 month questionnaire for clinicians: * On a scale of 1-5 (Strongly agree to

	Strongly disagree) I am confident providing smoking cessation assistance to pregnant women (self-efficacy²); * On a scale of 1-5 (Strongly agree to Strongly disagree) I am confident arranging follow up support for pregnant smokers (self-efficacy²);
Data analysis – I suggest the authors remain open to other measures of central tendency other than the mean in case distributions of scores are not normal – the median is usually preferable (if data is normally distributed it will align with the mean, if not, it will be a more valid indicator of central tendency). The qualitative analysis section was brief and did not really describe how themes would be generated.	Thank you this is helpful, we have adjusted 'mean' to 'measure of central tendency' in this section: However, where appropriate descriptive statistics (measures of central tendency, standard deviations and proportions) will be produced ...Analyses for the moderators will include calculation of a measure of central tendency, ... A measure of central tendency for each set of items that load onto the relevant subscale will be calculated for each subscale. A measure of central tendency of the scale scores will be calculated which will provide a total score for the Implementation Leadership Scale...(p19)
Figure 3 needs some re-formatting (probably the process of converting to pdf), though is a helpful overview.	Agreed – it looks fine on the version we submitted but agree that in conversion to pdf it seems to now need re-formatting
The references need some re-formatting too as some of the organisation names/authors haven't come across properly (I have the same issues with my reference management software and have to hand correct them!).	Thank you – we will pick this up at the copy editing stage, if the paper is accepted for publication.
After the reference section some of the tables are repeated but without titles.	We hope we have fixed this issue in our revised manuscript which has all figures as separate uploads

Reviewer: 2

Dr. Annariina Koivu, Tampere Universities

Reviewers' comment (NB page numbers from the marked-up Pdf provided by the reviewer)	Response (page numbers from the revised word document)
ABSTRACT:	

Spell out acronym of NSW in abstract.	Amended
Re "leaders" - I feel that this needs at least one more word to define the "leaders". It is explained later in the article but the abstract only should already provide a clear understanding.	We have added in a couple of examples of 'leaders': ...health system, leader (including managers and educators) and clinician components.
Strengths and limitations box needs reformatting	Thanks – we will pick this up at copy editing if the paper is accepted
INTRODUCTION:	
Re second sentence: I believe this first part of sentence refers to Australia, not generally or globally? Because it would not be the most important modifiable risk factor globally, but one among the other nutritional, environmental and infection related risk factors, see e.g. https://www.sciencedirect.com/science/article/pii/S0002916523397533?via%3Dihub Regarding Australia, if I was not 100% sure that it really is THE most important modifiable risk factor for adverse birth outcomes, I would say that it is among the most important risk factors or something like that. Why mention SGA, and not low birth weight, for which there has been consistent evidence for a long time?	Thank you for requesting clarification on this. We have altered the wording and added a citation, and have changed 'small for gestational age' to 'low birth weight' as shown in the extract below: Smoking in pregnancy is associated with a multitude of adverse outcomes for both mother and baby including pre-term birth and low birth weight babies.²⁻⁵ In Australia,

	smoking is the most common modifiable risk for adverse pregnancy and birth outcomes⁶... (p4 first paragraph)
Re NSW guidelines: Also WHO guidelines	We are aware of the WHO guidelines but at this point in the paragraph we are aiming to narrow the focus in to NSW specifically
Re Figure 1: Glossary = an alphabetical list of words relating to a specific subject. Is this the best word? With "glossary", I would expect to see other terms used in article, not just the description of participant target groups. Also, you do not explain what you mean by "systems" although you say so when you refer to this table.	We have changed the legend for Figure 1 and added an example of systems: It focuses on changing behaviours by targeting systems such as the electronic medical record system... (p4 second paragraph)
Re sentence beginning "midwifery educators" p6 of 42: Very full sentence, could almost work better as a list, table or figure.	We have separated the 4 points which are the main components of MOHMQuit out into a list as suggested
Re use of bracket p6 of 42: You have a plenty of important detail in brackets throughout this article. I think that for the most part they could be revised. Either leave out or structure the sentences and text differently, utilising tables, figures,	We have reviewed the manuscript and significantly

timelines and lists when possible.	reduced our use of brackets.
Re “Unlike many earlier interventions aimed at improving SCS” p6 of 42: Maybe this claim would benefit from a reference	We have added a reference to support this claim: Unlike many earlier interventions aimed at improving SCS,³...(p5 paragraph beginning MOHMQuit)
Re “components/strategies” p6 of 42: maybe pick one or the other	We have changed this to components
METHODS AND ANALYSIS:	
Re RE-AIM p7 of 42: please open	Added.
Re “but ‘Reach’ from the RE-AIM framework was specifically added...” p7 of 42: I appreciate that most of the readers will be expert audience, but this text written like that assumes that the reader knows already a lot of these frameworks. Please elaborate. Readers can be not only scientists but health professionals or health policy makers and these concepts should be clarified in a way that the reader does not need to start digging deeper to understand the main points.	Thank you for this feedback. We have elaborated to make this clearer: The Proctor implementation outcomes generally map on to other well-used frameworks such as the RE-AIM (Reach, Efficacy, Adoption, Implementation, Maintenance) framework but ‘Reach’ from the RE-AIM framework was

	specifically added into the design as 'Reach' captures the number of clinicians and leaders invited to and taking part in the trial. (p6 second paragraph)
Re "observations" p7 of 42: Observations are unclear - is this that you used observation as a data collection method (in which case it would need more details) or something else. This whole paragraph is slightly vagueish.	We have reworded this to avoid confusion. Subsequently, instruments were developed which encompassed both individual and service level data collection. (p6 third paragraph)
Re quote from Moore 2015 p7 of 42: I think this does not add much value, it is a bit generic statement that would fit in all kinds of projects.	We have retained this quotation as it makes the important point that process evaluations of complex interventions cannot provide answers to every question and this approach of aiming to answer key questions comprehensively and with rigour, guided our thinking
Re paragraph beginning "In part these foci..." p8 of 42: This is not a major point	Given this is a

but a matter of taste: I feel that these sentences are written a bit backwards, Important pieces of information, such as that you build on the learnings from the acceptability trial, or the the trial was short in duration (with important details brackets again!) sme how come as a secondary, additional clauses, whereas I would report them first.	matter of taste we have retained the original structure of this paragraph
Re Figure 2: A good figure is self-explanatory even without the body-text. Maybe this can be revised (including the figure title)? Now it looks a bit like an unfinished table, unless you read the actual article.	We have provided a bit more explanation and an example in the text (p7 final sentence) and have revised the title of Figure 2 to Speculating which context elements may impact on each of the implementation outcomes to make the figure more self-explanatory
RECRUITMENT AND CONSENT:	
Re sentence beginning “Individual service leaders and clinicians...” p10 of 42: Please add some (expected) N numbers for the reader to better understand the scale, context and quality of this study.	We have added a very approximate n into the detail of the section on Data collection from leaders and antenatal care clinicians (p9-10)
DATA ANALYSIS:	
Re sentence “At this stage it is not possible...” p22 of 42: I think that this is a risky sentence, because it can be read in a way that you will publish only positive outcomes etc or leave out something that does not fit the picture.	Thank you – we certainly did not intend for the sentence to be read that we would only publish positive outcomes. We will not exclude results because they ‘do not fit

	the picture', rather we will not focus on implementation outcomes where we are unable to draw any conclusions due to a lack of variation across sites e.g. if all sites have the same degree of fidelity.
Re sentence "However, where appropriate descriptive statistics..." p22 of 42: It is difficult to analyse the appropriateness of this because of missing N numbers	We have added a very approximate n into the detail of the section on Data collection from leaders and antenatal care clinicians (p9-10)
Re "triangulation" p22 of 42: An example could be helpful	We have provided an example: Data from multiple sources will facilitate triangulation, for example collecting data about acceptability from quantitative data (post-training questionnaires from clinicians and questionnaires at six months from all clinicians) along with qualitative interviews with leaders from

	each site. (p20 final paragraph)
Re Figure 3 “Qualitative data sources”: needs fixing up	Thank you this has been an issue with formatting which we will address at the next stage of this paper, if it progresses
DISCUSSION:	
Re sentence “For example, if it transpires...” p25 of 42: If your plan is to use measures of association (referring to a wide variety of statistics that quantify the strength and direction of the relationship between exposure and outcome variables, enabling comparison between different groups), this should have come earlier and more detailed manner. This is the first instance where you speak about association.	The sentence has been reworded: For example, if it transpires that implementation leadership is more evident in those sites where MOHMQuit was shown to be particularly effective... (p22 penultimate paragraph)
Re sentence “Whilst we have described...” p25 of 42: Vaping and the other ways of nicotine use are increasing whereas smoking is decreasing in many parts of the world. Maybe in your future Results article you can possibly discuss to what extent the learnings from the current study can be helpful in studying and implementing interventions addressing other nicotine products use.	Thank you for this suggestion about future papers.
STRENGTHS AND LIMITATIONS:	
Re sentence “However, MOHMQuit is a complex intervention...” p26 of 42: I am surprised that evaluation components that would somehow assess the mitigation of drop-out rates are not part of this study -maybe they are discussed elsewhere.	The MOHMQuit intervention targets antenatal care providers and leaders, involving them in a one-off training event and subsequent MOHMQuit

	components such as leadership support for smoking cessation. As such drop out rates are not a key part of this study in the way that they would be in other studies.
Re sentence "This may compromise..." p26 of 42: full stop missing	Thank you we have amended the sentence.

VERSION 2 – REVIEW

REVIEWER	Felix Naughton University of East Anglia Faculty of Medicine and Health Sciences, School of Health Sciences
REVIEW RETURNED	20-Feb-2024

GENERAL COMMENTS	I am happy with the authors' corrections though they seemed to have not addressed out one of my comments "The qualitative analysis section was brief and did not really describe how themes would be generated"
---

REVIEWER	Annariina Koivu Tampere Universities
REVIEW RETURNED	26-Feb-2024

GENERAL COMMENTS	The comments from the first review round have been adequately addressed. Minor revision is required to correct the remaining grammatical errors, such as those on lines 33 and 47.
--